# Ultracompact MXene V_2_C-Improved Temperature Sensor by a Runway-Type Microfiber Knot Resonator

**DOI:** 10.3390/nano13162354

**Published:** 2023-08-17

**Authors:** Si Chen, Junhong Ran, Tong Zheng, Qing Wu

**Affiliations:** 1School of Physics and Electronic Information, Gannan Normal University, Ganzhou 341000, China; chensics9@163.com; 2Heilongjiang Province Key Laboratory of Laser Spectroscopy Technology and Application, Harbin University of Science and Technology, Harbin 150080, China; ran_jh@126.com; 3School of Artificial Intelligence, Beijing Technology and Business University, Beijing 100048, China

**Keywords:** temperature sensor, MXene V_2_C, runway-type microfiber knot resonator, photothermal conversion efficiency

## Abstract

We demonstrate an all-fiber, compact-structure, high-sensing-efficiency temperature sensor using a resonator structure sensor device of a runway type and MXene V_2_C. The high-quality functional material MXene V_2_C, synthesized by a simple two-step method, has excellent photothermal conversion performance. As-prepared MXene V_2_C is integrated into the runway section of a runway-type microfiber knot resonator based on the coupling mechanism between the surface near the field of the fiber and materials. When the temperature variation range is ~25–70 °C, the corresponding transmission light intensity variation is linear, and the maximum normalized sensing efficiency is 2.21 dB/°C/mm. Our work demonstrates that the runway-type structure ensures the compactness of the sensor device and enhances the interaction distance between the material and the microfiber, which provides additional integration strategies for functional material-based sensor devices.

## 1. Introduction

Optical fiber sensors with the merits of electromagnetic resilience, high integration ability, chemical corrosion resistance, and remote sensing ability have received a lot of attention [1,2]. All-fiber interferometer and resonator structures are commonly used in the sensor field. However, the environmental instability of all-fiber sensor devices based on interferometer structures (multi-devices) limits their commercial availability. Microfiber has a strong evanescent field, low transmission loss, and outstanding flexibility, making it an essential component in the all-fiber sensor field [3]. Microfiber knot resonators (MKRs) are typically produced using micro/nanofibers, which can be easily obtained from single-mode fibers [4]. MKRs, resonator-based sensor devices, have better stability compared with microfiber coil resonators and microfiber ring resonators [5]. The strong evanescent field facilitates the role of fiber and functional materials [6,7,8], thereby improving sensing efficiency [9,10].

Two-dimensional (2D) materials, such as reduced graphene oxide (rGO) [11,12], graphene [13,14], antimonene [15,16], phosphorene [17,18], and MXene [19,20,21], have been used as functional materials in the field of sensors, especially in the field of temperature sensors. Benefits from the thermal application of 2D materials, combined with different sensing structures, including typical interferometer structures, such as Mach-Zehnder interferometer (MZI) and Michelson interferometer (MI) platforms, were earlier applied to all-fiber temperature sensors. However, the practical application of these interferometer-type sensor devices is restricted by their insufficient interference immunity and unsatisfactory sensing efficiency [20]. The temperature sensor device-resonator has been found to have superior sensing efficiency and environmental stability. The enhanced interaction distance between the functional material and fibers in the runway-type MKR contributes to its excellent resonance characteristics [14,20,21]. The choice of functional materials is also crucial. MXene, with the advantage of high thermal conductivity and photothermal conversion efficiency (~100%) [22,23,24], makes it an ideal material for sensors. A new MXene compound, vanadium carbide (V_2_C), has been reported to have excellent thermal and optical properties [25]. Although the potential of V_2_C MXene for thermal management is still in its early stages, exploring all-optical temperature sensor devices based on MXene V_2_C is highly promising [26].

Here, we report an all-fiber ultracompact temperature sensor integrating MXene V_2_C fabricated by a simple two-step method. The optical deposition method is used to control the distance between MXene V_2_C and the microfiber of the runway section. The spectral response of sensor devices (with and without MXene) is investigated (resonance parameters). The maximum sensing efficiency of ~0.32 dB/°C (normalized sensing efficiency of 2.21 dB/°C/mm) is observed in this MXene-based temperature sensor. Additionally, the maximum slope of the extinction ratio decrease against temperature is ~−0.073 dB/°C. Compared with ref. [27,28,29,30,31], this work demonstrates a significant advantage. The experimental results we obtained provide new strategies for combining MXene materials with all-fiber devices and applying them to the sensing field.

## 2. Device Fabrication and Materials Characterization

The V_2_C nanosheets are prepared by a simple two-step method (chemical etching process and ultrasonic stripping process). The etching process uses HF solution to remove the Al atoms from the MXene precursor V_2_AlC to obtain V_2_C powder. Then, as-prepared powder obtained by the etching process is added to the NMP solution and sonicated for 20 h. The NMP solution is centrifuged at 3000 rpm for 20 min, the supernatant is retained, and the precipitate is removed to remove the V_2_C that has not been stripped. The supernatant is then centrifuged at 18,000 rpm for 30 min and then removed and re-dispersed in the IPO solution for later use.

Transmission electron microscopy (TEM), Raman spectrum, and UV-Vis-NIR absorption spectroscopy are carried out to evaluate the morphological and spectroscopic features of V_2_C nanosheets. As shown in Figure 1a, the TEM image shows that V_2_C is a typical 2D slice structure with a lateral size of about 780–1700 nm, indicating that V_2_C nanosheets have been successfully prepared. To study the structural properties of V_2_C nanosheets, the Raman spectrum of the nanosheets has been investigated. As shown in Figure 1b, the Raman spectrum demonstrates several characteristic peaks of V_2_C nanosheets located at 138.9, 281.9, 405.5, 519.6, and 684.3 cm^−1^, respectively. V_2_AlC contains a Raman characteristic peak at ~256 cm^−1^, and the intensity of this characteristic peak gradually decreases with the increase in etching time until it is completely small; meanwhile, a new characteristic peak appears at ~281.9 cm^−1^, which corresponds to the E_1g_ vibrational mode [32]. The signal at this position may be influenced by the increase in layer spacing during V_2_C fabrication. According to previous work, the characteristic peak at ~404 cm^−1^ is from V_2_C(OH)_2_, while the peaks at 519.6 cm^−1^ and 684.3 cm^−1^ are from the in-plane A_1g_ vibrations of the V-atom model of V_2_CF_2_ and V_2_CO(OH), respectively [32,33,34]. The UV-Vis-NIR absorption spectrum of V_2_C nanosheets is shown in Figure 1c, which demonstrates outstanding broadband absorption characteristics in the range of 300–2000 nm.

The diameter of the microfibers used to fabricate runway-type MKR is ~6–10 μm, which balances the majority of evanescent light with light transmission power. The appropriate diameter of microfiber ensures that evanescent light leaks outside of the microfiber, facilitating the interaction between light and functional materials. As shown in Figure 2a, the microfibers (the preparation loss of ~0.1 dB/~0.08 dB at 1550/980 nm, the diameter of ~7.1 μm) used to form the MKR are created through the hot flame conical stretching method [4] using SMF-28 (Corning, Corning, NY, USA). The microscope image (red light) of the bare runway-type MKR is shown in Figure 2a with a long/short axis diameter of ~17.2/2.8 mm. To fabricate the MKR, the unstripped coating layer is gripped at both ends of the fiber, elevating it into mid-air and forming a large ring by tying a knot in the tapered region while simultaneously pulling the fiber tails to form a small D-shaped ring. As the ring’s diameter is reduced, the shape of the fiber ring changes from D-shape to elliptical to achieve the final device shape. It is essential to note that the microfiber’s diameter is exceptionally small, making it prone to bending and breaking. Therefore, during the pulling process, we avoid touching the microfiber portion and reduce the ring’s radius by tugging the uncoated fiber ends. This approach also prevents microfiber contamination. The microfiber is assembled into an intertwined MKR using a panning table and microscope. The resulting MKR package is then fixed onto a slide for further analysis.

The optical response measurement system of bare runway-type MKR is shown in Figure 3. The sensor light source is switched to amplified spontaneous emission (ASE, Shanghai, China, OS321752), and the output characteristics are tested by the YOKOGAWA optical spectrum analyzer (OSA, Musashino, Japan, AQ6307C). The measured transmission spectrum of the bare all-fiber resonator is presented in Figure 2b, which reveals key parameters at 1544.7 nm, including a free spectral range (*FSR*) of ~3.8 nm, a resonance Q-factor of ~1716.3, and a maximum extinction ratio (ER) of ~11.74 dB. The bare all-fiber resonator (runway-type MKR without MXene) has a good resonance effect, which is beneficial for its application in sensing.

## 3. Results and Discussion

As-prepared MXene is deposited onto the surface of the microfiber by optical deposition method (as shown in the inset of Figure 3), and the optical properties of the all-fiber sensor device (runway-type resonator) based on MXene are tested. The experimental setup for temperature sensing is shown in Figure 3. The temperature sensing is verified by inserting the slides containing the MXene device into the electric heating constant temperature incubator (DHP9042, Labonce, Beijing, China), and the sensing performance is tested at 5 °C intervals within the temperature range of 25–70 °C.

Figure 4a displays the transmission spectra of ultracompact sensor devices with and without V_2_C at a room temperature of ~25 °C. The two curves exhibit (a red solid line without V_2_C and a blue solid line with V_2_C) several noticeable differences. The ultracompact sensor device with V_2_C (blue) shows a decrease of ~6.0 dB in transmission compared with the ultracompact sensor device without V_2_C (red). This decrease is due to additional loss introduced through the V_2_C deposition process. The ultracompact sensor device with the V_2_C material (blue) has resonance parameters at 1534.2 nm, including a free spectral range (*FSR*) of ~3.9 nm, a resonance Q-factor of ~1917.8, and a maximum ER of ~14.1 dB. Furthermore, within the wavelength of 1537–1540 nm, the transmission spectrum of the ultracompact sensor device with V_2_C shows a smoother profile compared with the ultracompact sensor device without V_2_C. This indicates that only one primary resonance condition is met in the ultracompact sensor device with V_2_C, and other possible resonances are suppressed. In contrast, several resonances occurred in the ultracompact sensor device without V_2_C. Some minor resonances are visible as small depressions in the range, such as the one highlighted by the black dashed rectangular box in Figure 4a [35].

The output transmission spectra are recorded by the ultracompact sensor device without V_2_C when the temperature is increased from 25 °C to 70 °C at 1544.7 nm in Figure 4b. Analysis of the data reveals an amplitude variation Δ*T* of approximately 0.7 dB at the resonant dip wavelength *λ*_res_. However, no significant shift is observed in *λ*_res_, and almost no shift is found in *λ*_res_. The temperature sensitivity corresponding to these results is 0–0.02 dB/°C. These findings suggest that the MKR made of silica-based microfiber alone cannot achieve a better realization of optical amplitude tuning. The functional material MXene V_2_C contributes more to the sensing efficiency.

The performance index shows a significant difference between the ultracompact sensor device with and without V_2_C. Table 1 summarizes the resonance characteristics observed in runway-type MKR with and without V_2_C. For the runway-type MKR without V_2_C, the resonance wavelength *λ*_res_ with the maximum ER (11.74 dB) is 1544.7 nm. However, for the runway-type MKR with V_2_C, the *λ*_res_ of the maximum ER (~14.1 dB) occurs at a much smaller wavelength of 1534.2 nm. This difference in *λ*_res_ could be attributed to the variation in resonance order and its mode effective index [36].

The results presented in Table 1 show that the *FSR* of the ultracompact sensor device without V_2_C is 3.8 nm, while the *FSR* of the device with V_2_C is 3.9 nm. These findings confirm that the MXene did not affect the *FSR* of the sensor structure. When light passes through a microfiber with a diameter of approximately 7.1 μm, multiple modes are generated in the transition zone. Our study considers these modes during propagation, as shown in Figure 5. As a result, the larger *FSR* phenomenon cannot be solely attributed to the expansion of the bit phase difference. The constant distance (l) between the multiple modes remains unchanged. As a result, the impact of Δl in Equation (1) is often negligible, rendering the denominator in Equation (1) as $\Delta n_{eff}l$ only. Furthermore, changes in the refractive index ($\Delta n_{eff}l$) occur in the first-order, second-order, and third-order modes during passage through the transition zone. However, the change in $\Delta n_{eff}l$ is typically insignificant compared with other influential factors, resulting in a negligible $\Delta n_{eff}l$, which consequently leads to a larger *FSR*. This augmented *FSR* holds promising implications for the fabrication of all-fiber ultracompact sensor devices.
(1)$$\mathrm{FSR}=\frac{\lambda^{2}}{\Delta n_{eff}l+n_{eff}\Delta l}$$
where λ is the incident wavelength; $n_{eff}$ is the effective refractive index; and l is the perimeter of the runway section of the device.

The fundamental mode of the SMF (${\mathrm{HE}}_{11}$ mode) is excited in the transition region (Figure 5) to a higher order mode (${\mathrm{HE}}_{1n}$ mode, n≥2), and in the coupling region of the all-fiber sensor device, the three modes will independently cycle through the coupling region and then produce resonance effects [37]. Equation (2) is as follows [38]:(2)$$I_{\mathrm{output}}=(1-{ϒ}_{0})\delta \sum_{1}^{3}I_{\mathrm{MKR}-{\mathrm{HE}}_{1n}}+(1-\delta )I_{\mathrm{device}}$$
where ${ϒ}_{0}$ is coupling intensity loss, δ is interference factor, $I_{\text{E-MKR}-{\mathrm{HE}}_{1n}}$ is resonance intensity of ${\mathrm{HE}}_{1n}$ mode. $I_{\mathrm{device}}$ is the interference intensity [38].

MKR is a stable resonant structure achieved through strong evanescent field couplings. Light coupled into the ring satisfies the resonance conditions, while light coupled out of the ring interferes with the transmitted beam, forming a stable resonance spectrum. Changes in the surrounding temperature cause thermal expansion of the microfibers and their surface materials, resulting in variations in the microfiber’s length, core’s refractive index, and the refractive index of the external cladding materials. Consequently, the effective refractive index of the transmitted mode is altered, leading to significant changes in the transmitted power of the MKR.

The ultracompact sensor device with V_2_C is subjected to resonant amplitude tuning using the identical experimental setup (Figure 3). By inserting the slides containing the ultracompact sensor device with V_2_C into the electric heating constant temperature incubator (DHP9042, China), temperature sensing is verified at 5 °C intervals within the temperature range of 25–70 °C. The results show that the maximum Δ*T* is ~14.42 dB with a *λ*_res_ shift of ~0.3 nm at *λ* of 1534.1 nm at 25–70 °C (the inset of Figure 6a). Similarly, the second largest Δ*T* is ~13 dB at 1566.8 nm and 25–70 °C, with a *λ*_res_ shift of approximately 0.2 nm (the inset of Figure 6b).

To investigate the sensing characteristics at 1530–1570 nm, the fit of the experimental data in Figure 7a show that the amplitude variation (Δ*T*) with temperature is linear. Additionally, Figure 7b displays the extinction ratio versus temperature. To summarize our results, Table 2 provides information on the properties of the resonances, the associated Δ*T* rates of change, and the slope of the extinction ratio reduction against temperature at different resonance wavelengths.

The maximum sensing efficiency of the ultracompact sensor device with V_2_C is ~0.32 dB/°C. This efficiency is represented in Figure 7a (green) and is achieved at 1534.1 nm with the largest ER (~14.1 dB) (Table 2). The second largest sensing efficiency (0.294 dB/°C) of the ultracompact sensor device with V_2_C in Figure 7a (orange) with a smaller ER (~13.1 dB) at *λ*_res_ is 1566.8 nm (Table 2). There is a linear correlation between the extinction ratio of MKR and temperature, with a smaller extinction ratio observed at higher temperatures. Figure 7b illustrates the extinction ratio as a function of temperature, with the maximum slope of the extinction ratio decreasing against temperature at ~−0.073 dB/°C. This slope is obtained at the resonance wavelength *λ*_res_ of 1534.1 nm, corresponding to the first row in Table 2. The reduction in ER indicates that the strong absorption property of V_2_C will lead to the excitation of electron-hole pairs in the V_2_C nanosheets as the temperature increases. The carriers generated from these photons will lead to a change in the real and imaginary parts of the refractive index in the V_2_C nanosheets. The real part of the refractive index change in the V_2_C nanosheet is related to the wavelength shift of the resonance wavelength and manifests as a wave shift of the resonance wavelength. The change in the imaginary part of the refractive index in the V_2_C nanosheet may lead to a change in the resonance condition. Meanwhile, the concentration of photon excitation carriers increases with the increase in temperature. This will lead to an increase in the coupling loss factor of the resonator structure. The increase in the coupling loss factor will cause the resonant state to deviate from the critical coupling [39]. Therefore, the resonant ER can be found to decrease with increasing temperature near the resonant wavelength.

To ensure the stability of the ultracompact sensor device with V_2_C, we conducted rigorous experimental tests. The MXene-MKR sensor is placed within the electric heating constant temperature incubator and subjected to controlled environmental conditions with a fixed temperature of 25 °C and 70 °C. We monitored the output spectra for 120 min (at ten-minute intervals) and observed no significant intensity fluctuations at four wavelengths (Figure 8a,b). Figure 8 indicates that the intensity fluctuations are less than 1 dB, which is significantly smaller than the intensity variations caused by temperature changes at fixed wavelengths. These results indicate that the proposed runway-type resonator operated stably with the experimental conditions.

MKRs have demonstrated remarkable potential as highly sensitive optical sensors, attributed to their compact structure and strong evanescent field interactions with the surrounding environment. In our subsequent study, we conducted an in-depth exploration of the impact of microfibers of diameters on the sensor’s sensing efficiency. We investigated sensitivity variations for different microfiber diameters (~5, ~6, ~7, ~8, and ~9 µm) at an MXene V_2_C concentration of ~8 mg/mL (deposition length of ~145 μm) while maintaining similar MKR ring lengths. The sensing characteristics of MKRs with different microfibers of diameters are presented in Table 3.

The experimental results presented in Table 3 demonstrate that the MKR sensitivity reaches a maximum value of 0.32 dB/°C when the diameter of the microfibers decreases from ~9 µm to ~7 µm. This trend indicates that smaller-diameter microfibers offer enhanced sensitivity compared with larger diameters. The observed sensitivity variation in MKRs with different microfiber diameters can be attributed to the increased evanescent field interactions and stronger confinement of light within the smaller-diameter microfibers. Smaller-diameter microfibers provide a larger surface area per unit length, leading to enhanced interactions with the external medium, resulting in a higher sensitivity. Additionally, the increased confinement of light in smaller-diameter microfibers enhances the light–matter interactions, resulting in a stronger response to changes in the surrounding medium’s properties. However, when the diameter of the microfibers decreased from ~7 µm to ~5 µm, the sensitivity of the MKR did not continue to increase but rather decreased. It should be noted that while smaller-diameter microfibers can theoretically achieve higher sensitivity, they are also associated with higher optical loss. Therefore, in practical applications, careful consideration of the trade-off between sensitivity and optical loss is necessary when selecting the appropriate microfiber diameter for a specific sensor. Consequently, microfibers with a diameter of ~7 µm were selected for subsequent research in this study.

Next, we explore the sensing properties of MKR (microfiber with a diameter of ~7 µm) at varying concentrations of MXene V_2_C. The results are presented in Table 4.

Based on the experimental results, we observed a significant impact of MXene V_2_C concentration on the sensitivity of the MKR sensor. Increasing the MXene V_2_C concentration led to the following sensitivity trend: at ~2 mg/mL, the sensitivity was relatively low. At ~4 mg/mL, the sensitivity slightly increased but remained relatively low. Further increasing the MXene V_2_C concentration to ~6 mg/mL resulted in a significant sensitivity improvement. The highest sensitivity was achieved at an MXene V_2_C concentration of ~8 mg/mL. However, when the concentration reached ~10 mg/mL, the sensitivity of the MKR sensor began to decrease. This observed trend can be explained as follows: lower MXene V_2_C concentrations may not effectively enhance the interaction of the light field with the external environment, leading to lower sensitivity. On the contrary, gradually increasing the MXene V_2_C concentration enhances its coverage on the MKR surface, enabling stronger interactions with the external environment and thus improving sensitivity. Nevertheless, excessively high MXene V_2_C concentration may increase optical field loss and interference effects, causing a reduction in sensitivity.

Analysis of Table 3 and Table 4 highlights the substantial impact of microfiber diameter and deposited material concentration on the sensitivity of MKR. This study’s maximum normalized sensing efficiency (2.21 dB/°C/mm), as presented in Table 3, surpassed the previous findings of our group (1.65 dB/°C/mm) [40]. Hence, choosing the suitable microfiber diameter and deposition material concentration can notably enhance the performance of MKR sensors, providing heightened sensitivity and accuracy in specific application scenarios.

Table 5 shows the sensing characteristics of different types of all-fiber devices. The ultracompact sensor device with V_2_C demonstrated in this work outperforms the other configurations. Although the measurement temperature range of our work (25–70 °C) is moderate, there is still potential for improvement by further optimizing factors such as the area of the nanosheet coating and the diameter of the microfiber. Microfibers provide smaller diameters for MKRs and offer greater sensitivity per unit length of MKR than traditional fiber sensor devices with interferometric structures. This compact structure empowers MKRs to exhibit faster response times when interacting with the environment compared with other sensors. The runway-type MKR discussed in this paper has demonstrated heightened sensor sensitivity compared with the other all-fiber devices presented in Table 5. This improved sensing capability arises from the synergistic effect of the microfiber structure and MXene’s photothermal properties. The surface of the runway-type MKR is coated with MXene materials, which significantly enhances the interaction distance between the material and the fiber. Additionally, the high photothermal conversion efficiency and thermal conductivity of V_2_C materials contribute to the sensing performance of the all-fiber sensor device.

## 4. Conclusions

We have demonstrated an all-fiber, high-sensing-efficiency temperature sensor based on MXene V_2_C. The highest sensing efficiency of ~0.32 dB/°C (normalized sensing efficiency of 2.21 dB/°C/mm) is observed in the runway-type MKR coated with V_2_C, which is fabricated using 7.1 μm-diameter microfibers. This efficiency is achieved at *λ*_res_ of 1534.1 nm with a Q of ~1917.8 and an ER of ~14.1 dB. The runway structure used in our all-fiber sensor device significantly enhances the interaction length between light and the MXene V_2_C, thereby improving the overall sensing efficiency of the sensor. Simultaneously, the selection of appropriate microfiber diameter and MXene V_2_C concentration is crucial in achieving high sensitivity in MKR sensors. The experimental results underscore the significance of striking the right balance between the diameters of microfibers and deposited material concentration to optimize sensitivity while maintaining excellent optical performance. This device shows promise for developing fiber-compatible devices with functionalities.

## Figures and Tables

**Figure 1 nanomaterials-13-02354-f001:**
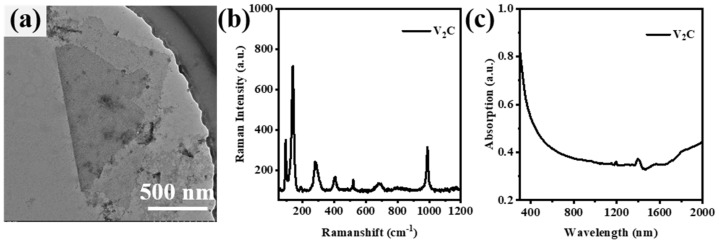
Characterization of V_2_C: (**a**) TEM of V_2_C nanosheets; (**b**) Raman spectrum of V_2_C; (**c**) broadband (300–2000 nm) absorption spectrum of V_2_C.

**Figure 2 nanomaterials-13-02354-f002:**
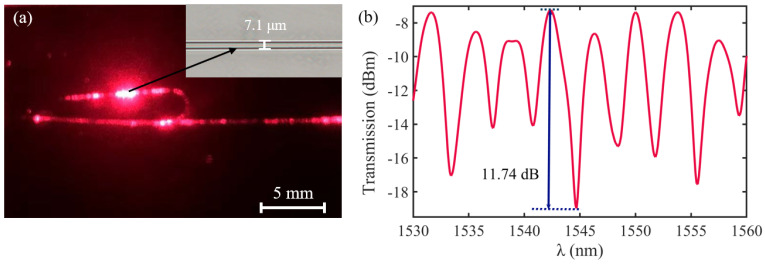
(**a**) Optical microscopic images of bare runway-type MKR illuminated by red laser (inset: the diameter of microfiber for preparing runway-type MKR is ~7.1 μm). (**b**) Transmission spectra.

**Figure 3 nanomaterials-13-02354-f003:**
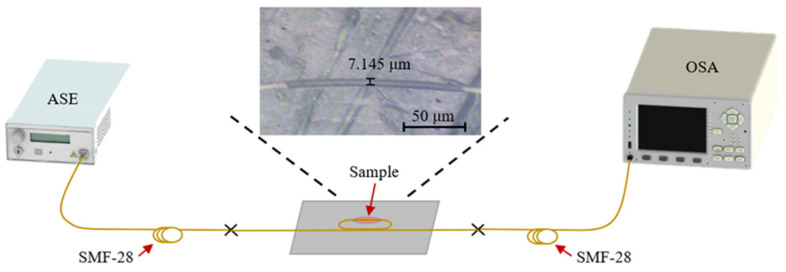
Experimental setup for temperature sensing.

**Figure 4 nanomaterials-13-02354-f004:**
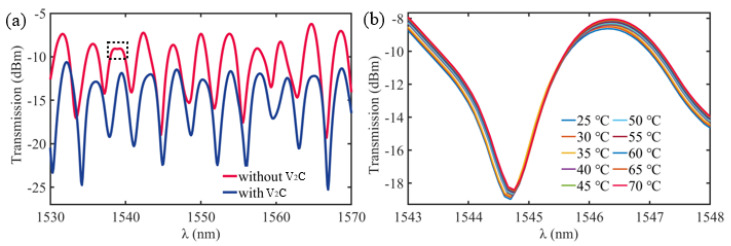
(**a**) Transmission spectra of the ultracompact sensor device without V_2_C (red) and with V_2_C (blue). (**b**) Transmission spectra of the ultracompact sensor device without V_2_C at different temperatures.

**Figure 5 nanomaterials-13-02354-f005:**
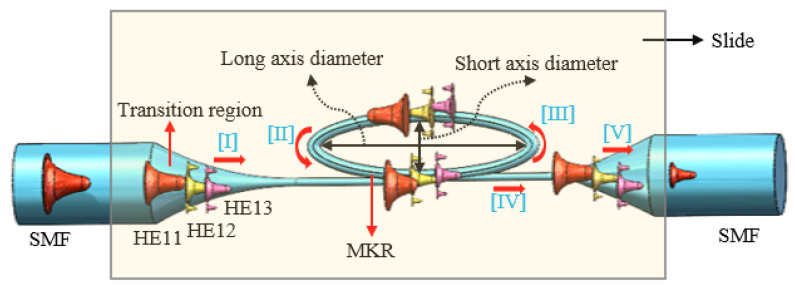
Schematic diagram of the all-fiber sensor device.

**Figure 6 nanomaterials-13-02354-f006:**
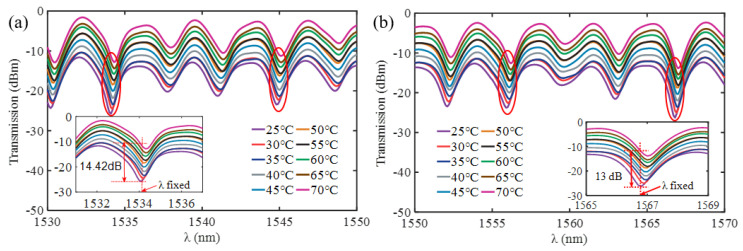
Transmission spectra of the ultracompact sensor device with V_2_C. (**a**) at 1530–1550 nm (**b**) at 1550–1570 nm.

**Figure 7 nanomaterials-13-02354-f007:**
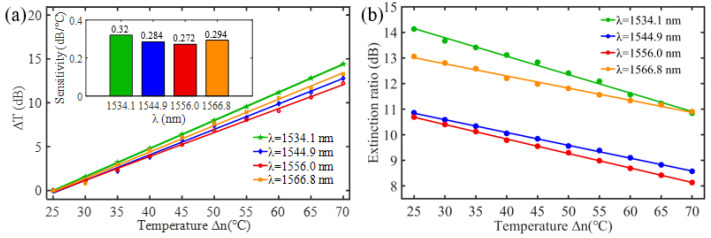
(**a**) Linear fit of Δ*T* vs. temperature. (**b**) The extinction ratio vs. temperature.

**Figure 8 nanomaterials-13-02354-f008:**
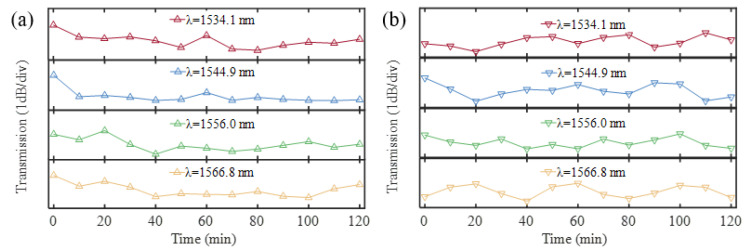
The output spectra. (**a**) at 25 °C. (**b**) at 70 °C.

**Table 1 nanomaterials-13-02354-t001:** Resonance properties of the ultracompact sensor device with and without V_2_C.

Structure	λres (nm)	ERmax (dB)	Q (10^−3^)	FSR (nm)
device without V_2_C	1544.7	11.74	1.7163	3.8
device with V_2_C	1534.2	14.1	1.9178	3.9

**Table 2 nanomaterials-13-02354-t002:** Sensing characteristics at different resonance wavelengths.

λ_res_ (nm)	ER at 25 °C (dB)	Δ*T* at 70 °C (dB)	Δ*T*/Δn (dB/°C)	ΔER/Δn (dB/°C)
1534.1	14.1	14.42	0.320	−0.073
1544.9	11.1	12.79	0.284	−0.051
1556.0	10.8	12.22	0.272	−0.056
1566.8	13.1	13.27	0.295	−0.047

**Table 3 nanomaterials-13-02354-t003:** Sensing characteristics of MKRs with different microfiber diameters (MXene V_2_C concentration of ~8 mg/mL).

Microfiber Diameter	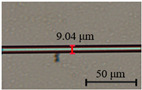	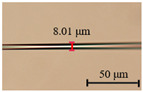	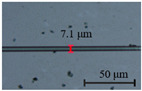	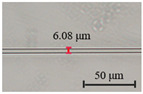	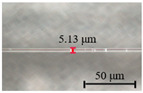
Δ*T* vs. temperature	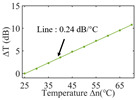	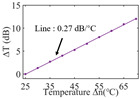	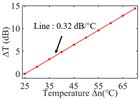	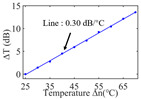	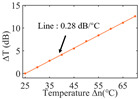
Sensitivity (dB/°C)	0.24	0.27	0.32	0.30	0.28
Normalized sensitivity (dB/°C/mm)	~1.66	~1.86	~2.21	~2.07	~1.93

**Table 4 nanomaterials-13-02354-t004:** The effect of different concentrations of MXene V_2_C on MKR transport properties with the same microfiber diameter (~7.0 µm)**.**

Different Concentrations	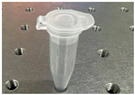	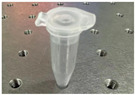	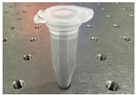	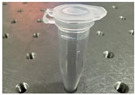	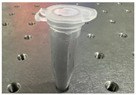
Concentration (mg/mL)	~2	~4	~6	~8	~10
Δ*T* vs. temperature	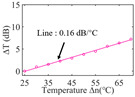	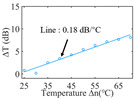	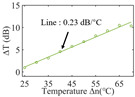	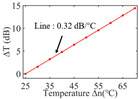	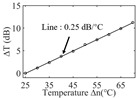
Sensitivity (dB/°C)	0.16	0.18	0.23	0.32	0.25
Normalized sensitivity (dB/°C/mm)	~1.10	~1.24	~1.59	~2.21	~1.72

**Table 5 nanomaterials-13-02354-t005:** Sensing characteristics of different types of all-fiber device structures.

Type of Structure	Sensitivity (dB/°C)	Temperature (°C)	Ref.
MF *^a^* with Graphene	0.03	20–75	[27]
MLR *^b^*	0.043	25–60	[28]
SPF *^c^* with TiO_2_	0.044	−7.8–77.6	[29]
MF with Graphene	0.1018	30–80	[30]
SPF with rGO *^d^*	0.134	−7.8–77	[31]
**Runway-type MKR + V_2_C**	**0.32**	**25–70**	**This work**

*^a^*: microfiber. *^b^*: microfiber loop resonator. *^c^*: side-polished fiber. ^d^: reduced graphene oxide.

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
