# Peer review of "Ultracompact MXene V2C-Improved Temperature Sensor by a Runway-Type Microfiber Knot Resonator"

_nanomaterials, 2023, doi:10.3390/nano13162354_

Round 1

Reviewer 1 Report

Reviewer Comments to the Authors

The present manuscript entitled “Ultracompact MXene V2C-improved temperature sensor by a 2 Runway-type Microfiber Knot Resonator”. The authors demonstrated for the first time, an all-fiber, high sensing efficiency (0.32 dB/ºC) temperature sensor based on MXene V2C. But, the authors failed to explain proper mechanism and scientific explanation for the high sensitivity of your temperature. Therefore, the reviewer recommends the manuscript in the Nanomaterials after major Revision.

Please see the below comments

1.    Sensitivity of your temperature sensor (0.32 dB/ºC) is very high when compared with other sensors.  But not explained proper mechanism in the manuscript. The authors must be explained mechanism and scientific explanation for the high sensitivity of your temperature sensor?

2.    Sensing temperature range is 25 ºC to 75 ºC. Why do you restrict the temperature range till only 75 C. Is there any specific purpose? Why you didn’t measure below 0 ºC?

3.    V2C nanosheet dimensions (length & width) are not included in TEM analysis. Please include them

4.    Please expand MKRs in line no:32 and remove the MKRs in line no.34.

Author Response

Comments from the reviewer 1:

  1. Sensitivity of your temperature sensor (0.32 dB/ºC) is very high when compared with other sensors. But not explained proper mechanism in the manuscript. The authors must be explained mechanism and scientific explanation for the high sensitivity of your temperature sensor?

Response: We acknowledge the reviewer for this comment. We have added the following sentence to the revised manuscript to further explain the temperature sensing mechanism. Thank you again for your comments.

“Although the measurement temperature range of our work (25-70 ℃) is a moderate, there is still potential for improvement by further optimizing factors such as the area of the nanosheet coating and the diameter of the microfiber. Microfibers provide smaller diameters for MKRs and offer greater sensitivity per unit length of MKR than traditional fiber sensor devices with interferometric structures. This compact structure empowers MKRs to exhibit faster response times when interacting with the environment compared to other sensors. The runway-type MKR discussed in this paper demonstrate heightened sensor sensitivity compared to the other all-fiber devices presented in Table 5. This improved sensing capability arises from the synergistic effect of the microfiber structure and MXene's photothermal properties. The surface of the runway-type MKR is coated with MXene materials, which significantly enhances the interaction distance between the material and the fiber. Additionally, the high photothermal conversion efficiency and thermal conductivity of V2C materials contribute to the sensing performance of the all-fiber sensor device.”

  1. Sensing temperature range is 25 ºC to 75 ºC. Why do you restrict the temperature range till only 75 ℃. Is there any specific purpose? Why you didn’t measure below 0 ºC?

Response: We acknowledge the reviewer for this valuable comment.

First, the purpose of the runway-type MKR temperature sensor, designed in this manuscript, is to monitor temperature changes in the general environment. Thus, we focus on its application range of 25-70°C.

Secondly, the sensitivity of this temperature sensor is significantly enhanced by the high thermal conductivity and photothermal conversion efficiency of the MXene V2C materials within the aforementioned temperature range.

Finally, the experimental temperatures in our laboratory are currently limited to the mentioned temperature interval in the manuscript. According to the valuable reviewer comments, we will study lower or higher temperatures to further expand the temperature range of the temperature sensor under study, especially in the study of sensing characteristics in extremely cold environments (regional environment).

  1. V2C nanosheet dimensions (length & width) are not included in TEM analysis. Please include them.

Response: We acknowledge the reviewer for this comment. We have made the following supplementary explanations in the revised manuscript. The lateral size of TEM image has been added in the revised manuscript.

“Transmission electron microscopy (TEM), Raman spectrum and UV-Vis-NIR absorption spectroscopy have been carried out to evaluate the morphological and spectroscopic features of V2C nanosheets. As shown in Figure 1a, the TEM image shows that V2C is a typical 2D slice structure with lateral size of about 780 nm-1700 nm, indicating that V2C nanosheets have been successfully prepared.”

  1. Please expand MKRs in line no:32 and remove the MKRs in line no.34.

Response: We acknowledge the reviewer for this comment. We agree with the reviewers and amend them as follows.

“However, the environmental instability of all-fiber sensor devices based on interferometer structures (multi-devices) limits their commercial availability. Microfiber, has strong evanescent field, low transmission loss, and outstanding flexibility, making it an essential component all-fiber sensor filed [3]. Microfiber knot resonators (MKRs) are typically produced using micro/nanofibers, which can be easily obtained from single-mode fibers [4]. MKRs, resonator-based sensor devices, have better stability compared to microfiber coil resonators and microfiber ring resonators [5].”

Reviewer 2 Report

The manuscript, “Ultracompact MXene V2C-improved temperature sensor by a Runway-type microfiber knot resonator” by Chen et al are designed and fabricated to a temperature sensor. A variety of techniques were used to characterize the obtained material, which demonstrated improved sensitivity. Also, I believe that this work is acceptable for publication in nanomaterials with the following minor improvements:

1.      The authors should properly discuss and describe the crystal planes in the XRD patterns of prepared catalyst.

2.      The Raman analysis should be discussed in detail by authors with the appropriate citations, as discussed in previous papers.

3.      The authors must reevaluate the UV-Vis spectrum and clearly explain their findings in the revised manuscript.

4.      The temperature sensing mechanism and conclusion section should be discussed in detail by the authors.

5.      Some of the important references need to cite in the revised manuscript. DOI: 10.1039/C6NJ04030F; 10.1002/elan.201900134.

Minor editing of English language required

Author Response

Comments from the reviewer 2:

  1. The authors should properly discuss and describe the crystal planes in the XRD patterns of prepared catalyst.

Response: We acknowledge the reviewer for this comment. Attributed to the high photothermal conversion efficiency and excellent thermal conductivity of V2C nanosheets, V2C is used in fiber type temperature sensors in this work.V2C is not used as a catalyst. In our manuscript, the photothermal properties of V2C nanosheets are mainly utilized, and their crystalline surface does not significantly affect the experimental results, so we did not investigate the XRD patterns of V2C in the overall study. If you insist that additional XRD data is needed, we will try to test it later.

This manuscript verifies the performance of the MXene from the point of view of device sensing performance.

First, in this work the microfiber is only used as a carrier of the sensor-device, and high efficiency sensing is achieved benefit from the coupling effect of the micro-nano structure and the photothermal properties of the MXene.

Secondly, in this experiment, the sensor performance of the runway type MXene-MKR and bare runway type MKR are compared, and the sensing efficiency of runway type MXene-MKR is significantly better than bare runway type MKR by analyzing and comparing their respective output spectra, which shows that the photothermal conversion efficiency of MXene is favorable for its application in the sensing field.

  1. The Raman analysis should be discussed in detail by authors with the appropriate citations, as discussed in previous papers.

Response: Thank you for your suggestion, more detail Raman analysis has been added in the revised manuscript.

“V2AlC contains a Raman characteristic peak at ~256 cm-1, and the intensity of this characteristic peak gradually decreases with the increase of etching time until it is completely small, meanwhile, a new characteristic peak appears at ~281.9 cm-1, which corresponds to the E1g vibrational mode [32]. The signal at this position may be influenced by the increase of layer spacing during V2C fabrication. According to previous work, the characteristic peak at ~404 cm-1 is from V2C(OH)2, while the peaks at 519.6 cm-1 and 684.3 cm-1 are from the in-plane A1g vibrations of the V-atom model of V2CF2 and V2CO(OH), respectively [32-35]. The UV-Vis-NIR absorption spectrum of V2C nanosheets is shown in Figure 1c, which demonstrate outstanding broadband absorption characteristics in the range of 300-2000 nm.”

  1. Guan, Y.; Jiang, S.; Cong, Y.; Wang, J.; Dong, Z.; Zhang, Q.; Yuan, G.; Li, Y.; Li, X. A hydrofluoric acid-free synthesis of 2D vanadium carbide (V2C) MXene for supercapacitor electrodes. 2D Materials 2020, 7, doi:10.1088/2053-1583/ab6706.
  2. Ghasali, E.; Orooji, Y.; Azarniya, A.; Alizadeh, M.; Kazem-zad, M.; TouradjEbadzadeh. Production of V2C MXene using a repetitive pattern of V2AlC MAX phase through microwave heating of Al-V2O5-C system. Applied Surface Science 2021, 542, doi:10.1016/j.apsusc.2020.148538.
  3. Champagne, A.; Shi, L.; Ouisse, T.; Hackens, B.; Charlier, J.-C. Electronic and vibrational properties ofV2C-based MXenes: From experiments to first-principles modeling. Physical Review B 2018, 97, doi:10.1103/PhysRevB.97.115439.
  4. Champagne, A.; Shi, L.; Ouisse, T.; Hackens, B.; Charlier, J.-C. Electronic and vibrational properties ofV2C-based MXenes: From experiments to first-principles modeling. Physical Review B 2018, 97, 115439, doi:10.1103/PhysRevB.97.115439.

  1. The authors must reevaluate the UV-Vis spectrum and clearly explain their findings in the revised manuscript.

Response: Thank you for your suggestion. The absorption spectrum in the manuscript is obtained by measuring the V2C nanosheet IPO dispersion. The absorption spectrum in the figure seems to be a bit strange, which could be caused by the organic functional groups in the IPO solution and the change of light source in the instrument. Therefore, we have corrected the absorption spectrum and the revised figure is shown below.

  1. The temperature sensing mechanism and conclusion section should be discussed in detail by the authors.

Response: We thank the reviewer for this comment and have added the following sentence to the revised manuscript to further explain the temperature sensing mechanism. Thank you again for your comments.

“MKR is a stable resonant structure achieved through strong evanescent field couplings. Light coupled into the ring satisfies the resonance conditions, while light coupled out of the ring interferes with the transmitted beam, forming a stable resonance spectrum. Changes in the surrounding temperature cause thermal expansion of the microfibers and their surface materials, resulting in variations in the microfiber's length, core's refractive index, and the refractive index of the external cladding materials. Consequently, the effective refractive index of the transmitted mode is altered, leading to significant changes in the transmitted power of the MKR.”

“The runway structure used in our all-fiber sensor device significantly enhances the interaction length between light and the MXene V2C, thereby improving the overall sensing efficiency of the sensor. Simultaneously, the selection of appropriate microfiber diameter and MXene V2C con-centration is crucial in achieving high sensitivity in MKR sensors. The experimental results underscore the significance of striking the right balance between microfibers of diameters and deposited material concentration to optimize sensitivity while maintaining excellent optical performance. This device shows promise for developing fiber-compatible devices with functionalities.”

  1. Some of the important references need to cite in the revised manuscript. DOI: 10.1039/C6NJ04030F; 10.1002/elan.201900134.

Response: We acknowledge the reviewer for the comment. We strongly agree with the reviewer's comments and have added the appropriate references in the revised manuscript.

“Two-dimensional (2D) materials, as functional materials, such as reduced graphene oxide (rGO) [11,12], graphene [12,13], antimonene [14,15], phosphorene [16,17], MXene [18-21], have been used in the field of sensors, especially in the field of temperature sensors.”

References:

[11] Amala, G.; Saravanan, J.; Yoo, D.J.; Kim, A.R.; Kumar, G.G. An environmentally benign one pot green synthesis of reduced graphene oxide based composites for the enzyme free electrochemical detection of hydrogen peroxide. NEW JOURNAL OF CHEMISTRY 2017, 41, 4022-4030, doi:10.1039/c6nj04030f.

[12] Gabunada, J.C.; Vinothkannan, M.; Kim, D.H.; Kim, A.R.; Yoo, D.J. Magnetite Nanorods Stabilized by Polyaniline/Reduced Graphene Oxide as a Sensing Platform for Selective and Sensitive Non-enzymatic Hydrogen Peroxide Detection. ELECTROANALYSIS 2019, 31, 1524-1533, doi:10.1002/elan.201900134.

Reviewer 3 Report

Journal: Nanomaterials

Manuscript Title: Ultracompact MXene V2C-improved temperature sensor by a Runway-type Microfiber Knot Resonator

Manuscript ID: nanomaterials-2508425-peer-review-v1

Authors: Si Chen , Junhong Ran , Tong Zheng, Qing Wu

The objective of this paper is to detail the advancement of a temperature sensor-oriented microfiber knot resonator, enhanced with V2C-based MXene material.

The primary issue with the presented content is that the current paper seems to be a duplicate or replication of another study that has already been published by some of the same authors, focusing on an extremely similar or identical subject matter. This duplication raises a significant obstacle to the publication of the paper in Nanomaterials.

(https://pubs.rsc.org/en/content/articlelanding/2023/ra/d3ra03190j).

The authors are running similar experimental setups with small differences of the chosen parameters.

In conclusion they noticed that ”The highest sensing efficiency of ~ 0.32 dB/°C is observed in the runway-type MKR coated with V2C, which is fabricated using 7.1 μm-diameter microfibers.” while in their publication in RSC Advances they concluded that the highest sensing efficiency of ~ 0.33 dB/°C is observed in the runway-type MKR coated with V2C, which is fabricated using 8.1 μm-diameter microfibers.

Consequently, readers can anticipate forthcoming articles to explore sensing efficiencies around ~0.31 dB/°C, 0.34 dB/°C, and beyond.

In my view, the present paper is unsuitable for publication, not only in Nanomaterials but also in any other academic journal.

There are of course other critical remarks on the content of the manuscript, for example line 69, MXene powder is added in NMP, then is sonicated for 20 h. Then what?

Line 109, Figure 2 is presenting Microscope image of what?

Etc.

Journal: Nanomaterials

Manuscript Title: Ultracompact MXene V2C-improved temperature sensor by a Runway-type Microfiber Knot Resonator

Manuscript ID: nanomaterials-2508425-peer-review-v1

Authors: Si Chen , Junhong Ran , Tong Zheng, Qing Wu

The objective of this paper is to detail the advancement of a temperature sensor-oriented microfiber knot resonator, enhanced with V2C-based MXene material.

The primary issue with the presented content is that the current paper seems to be a duplicate or replication of another study that has already been published by some of the same authors, focusing on an extremely similar or identical subject matter. This duplication raises a significant obstacle to the publication of the paper in Nanomaterials.

(https://pubs.rsc.org/en/content/articlelanding/2023/ra/d3ra03190j).

The authors are running similar experimental setups with small differences of the chosen parameters.

In conclusion they noticed that ”The highest sensing efficiency of ~ 0.32 dB/°C is observed in the runway-type MKR coated with V2C, which is fabricated using 7.1 μm-diameter microfibers.” while in their publication in RSC Advances they concluded that the highest sensing efficiency of ~ 0.33 dB/°C is observed in the runway-type MKR coated with V2C, which is fabricated using 8.1 μm-diameter microfibers.

Consequently, readers can anticipate forthcoming articles to explore sensing efficiencies around ~0.31 dB/°C, 0.34 dB/°C, and beyond.

In my view, the present paper is unsuitable for publication, not only in Nanomaterials but also in any other academic journal.

There are of course other critical remarks on the content of the manuscript, for example line 69, MXene powder is added in NMP, then is sonicated for 20 h. Then what?

Line 109, Figure 2 is presenting Microscope image of what?

Etc.

Author Response

Comments from the reviewer 3:

  1. The primary issue with the presented content is that the current paper seems to be a duplicate or replication of another study that has already been published by some of the same authors, focusing on an extremely similar or identical subject matter. This duplication raises a significant obstacle to the publication of the paper in Nanomaterials.(https://pubs.rsc.org/en/content/articlelanding/2023/ra/d3ra03190j).The authors are running similar experimental setups with small differences of the chosen parameters.

Response: We acknowledge the reviewer for the comment. There is some continuity between the research content of this manuscript and the previous research results. We apologize for the lack of innovative pertinence in the writing of this manuscript.

Firstly, in terms of device preparation in this manuscript, we created a database of microfiber through a large amount of data, which contains a large number of parameters, such as microfiber diameter, tapered length, preparation loss, deposition power, deposition loss, etc. (1) For the preparation of saturable absorbers in mode-locked fiber lasers, the database shows that the mode-locked pulse output can be realized when the diameter of the microfiber is ~ 6μm, the preparation loss is 0.1dB@1550nm, and the deposition loss is 3dB@1550nm. (2) For the preparation of sensor devices in our manuscript, the database shows that when the diameter of microfiber is ~7μm, it is conducive to the deposition of materials and has excellent sensing parameters; when the diameter is lower than 7μm, the preparation loss is large, which affects the ER of the sensor; when the diameter is higher than 7μm, the evanescent field is low, which is not conducive to the deposition of materials.

For the work of microfiber database, we have included the table3 in the revised manuscript, hoping that more database contents can be summarized in the subsequent work.

Secondly, we start from the material concentration and test the characteristics of the sensor at different concentrations. In this manuscript, optical deposition is performed with a solution of 5μL. Both the thickness and length of the material deposition affect the performance of the device. The deposition power can control the length of the material deposition (deposition length of ∼145 μm in this work), but its thickness can be controlled according to the concentration of the solution. Therefore, we also made comparative measurement of solution concentration (2/4/6/8/10mg/mL) in Table 4. When the concentration is 8mg/mL, the maximum normalized sensing efficiency is 2.21 dB/°C/mm.

For the work of material concentration testing, we will consider quantum dots and 3D printing and other methods in the future, but the traditional way of this manuscript has a certain economy, and only needs to make it controllable through the database.

Compared with the previous work, this work is more systematic, the preparation of the device has high repeatability, and the goal is strong. According to the database parameters, it is very easy to realize the preparation of sensor parts with different properties.

Table 3. Sensing characteristics of MKRs with different microfiber diameters (MXene V2C concentration of ∼8 mg/mL).

Microfiber diameter

ΔT vs

temperature

Sensitivity (dB/°C)

0.24

0.27

0.32

0.30

0.28

Normalized

sensitivity (dB/°C/mm)

∼1.66

∼1.86

∼2.21

∼2.07

∼1.93

Table 4. The effect of different concentrations of MXene V2C on MKR transport properties with the same microfiber diameter (~7.0 µm).

different

concentrations

Concentration (mg/mL)

∼2

∼4

∼6

∼8

∼10

ΔT vs

temperature

Sensitivity (dB/°C)

0.16

0.18

0.23

0.32

0.25

Normalized

sensitivity (dB/°C/mm)

~1.10

~1.24

~1.59

~2.21

~1.72

  1. In conclusion they noticed that ”The highest sensing efficiency of ~ 0.32 dB/°C is observed in the runway-type MKR coated with V2C, which is fabricated using 7.1 μm-diameter microfibers.” while in their publication in RSC Advances they concluded that the highest sensing efficiency of ~ 0.33 dB/°C is observed in the runway-type MKR coated with V2C, which is fabricated using 8.1 μm-diameter microfibers. Consequently, readers can anticipate forthcoming articles to explore sensing efficiencies around ~0.31 dB/°C, 0.34 dB/°C, and beyond. In my view, the present paper is unsuitable for publication, not only in Nanomaterials but also in any other academic journal.

Response: We acknowledge the reviewer for the comment. We are very sorry that the parameters in the manuscript are not clearly stated.

We defined the sensing efficiency (dB/°C) corresponding to unit length as normalized sensing efficiency (dB/°C/mm). Therefore, the deposition length of this manuscript is 145μm, and the deposition length of ref [41] is 200μm. Due to the database created by us, through parameter optimization, when the diameter of microfiber is 7μm, the fabrication loss is 0.1dB/0.08dB@1550nm/980nm, the material concentration is 8mg/mL, and the volume of deposited material is 5μL, the optimal normalized sensing efficiency of 2.21dB /°C/mm is obtained.

The maximum normalized sensing efficiency in this manuscript is 2.21 dB/°C/mm, and the corresponding maximum normalized sensing efficiency in ref [41] is 1.65 dB/°C/mm. The parameters in this manuscript have been verified by a large number of repeated experiments.

We have included part of the database information (Table3 and Table4) and corresponding instructions in the revised manuscript.

The film thickness was characterized by step profiler (DektakXT) testing methods. As shown below, the thickness of the film is ~3μm. We will later add database information based on material thickness, making this traditional deposition method fully repeatable in terms of device preparation.

  1. There are of course other critical remarks on the content of the manuscript, for example line 69, MXene powder is added in NMP, then is sonicated for 20 h. Then what?

Response: We acknowledge the reviewer for the comment. We have added detailed instructions to the revised manuscript.

“Then, as-prepared powder obtained by etching process is added to the NMP solution and sonicated for 20 hours. The NMP solution is centrifuged at 3000 rpm for 20 minutes, the supernatant is retained and the precipitate is removed to remove the V2C that has not been stripped. The supernatant is then centrifuged at 18,000 rpm for 30 minutes, the supernatant is removed and re-dispersed in the IPO solution for later”

  1. Line 109, Figure 2 is presenting Microscope image of what?

Response: We acknowledge the reviewer for the comment. We have added detailed instructions to the revised manuscript. Thank you again for your valuable comments.

“Figure 2. (a) Optical microscopic images of bare runway-type MKR illuminated by red laser (inset: the diameter of microfiber for preparing runway structure MKR is ~7.1μm.)”

Round 2

Reviewer 1 Report

The authors addressed all the questions and therefore the reviewer recommends the journal article in Nanomaterials